# Microtubules as a versatile reference standard for expansion microscopy
Rajdeep Chowdhury[1,2,8], Tiago Mimoso[3,8], Abed Alrahman Chouaib[4], Nikolaos Mougios[1,5], Donatus Krah[1], Felipe Opazo [1,5,6], Sarah Köster [3,7 ✉], Silvio O. Rizzoli [1,5,7 ✉] & Ali H. Shaib [1 ✉]

Expansion microscopy (ExM) is continually improving, and new ExM variants need to be validated on well-defined biological structures. There is no consensus on validation structures for ExM, especially as nuclear pore complexes or DNA nanorulers are not popular for ExM studies. Here we propose that microtubules should be used for ExM validation. The diameter of microtubules immunostained using primary and secondary antibodies is sufficiently large for the validation of techniques with resolutions better than 50 nm. For techniques with higher precision (up to ~10 nm), microtubules can be assembled and imaged in vitro, using a protocol that we introduce here. Alternatively, a cellular extraction procedure can be employed, followed by labeling the peptide chains of the tubulin molecules with NHS-ester fluorophores. Finally, for nanometer-scale techniques, single tubulin molecules can be analyzed. We conclude that microtubules are valuable structures for the validation of ExM and related technologies.

The introduction of super-resolution fluorescence microscopy, two decades ago, revolutionized cell biology imaging, enabling the analysis of molecules and organelles at precisions approaching the electron microscopy domain[1]. Several super-resolution principles have been introduced, all aiming to circumvent one fundamental problem of conventional, diffraction-limited imaging: the simultaneous emission of fluorescence from closely-spaced fluorophores. This problem was initially solved by addressing the fluorophore behavior, using patterned light beams that impose the coordinates from which emission from fluorescent molecules is permitted, as in the stimulated emission depletion microscopy family (STED)[2], and the saturated structured illumination microscopy family (SIM)[3]. Single-molecule localization microscopy (SMLM) followed a related path, by analyzing the positions of fluorophores that are allowed to emit randomly, e.g.,[4]. An alternative path was followed later by expansion microscopy (ExM)[5], which does not affect the simultaneous emission of fluorophores, but rather their spacing, by enlarging the sample after embedding in a swellable gel.

All of these technologies have been employed on a multitude of biological samples, from entire *Drosophila* larvae to organelles isolated from neurons or cancer cells, but their calibration at the nanometer level has typically relied on a handful of structures that are well understood and universally accepted as molecular rulers. Such samples include microtubules, nuclear pore complexes (NPC)[6], and, to a lesser extent, mitochondria, e.g.,[7,8]. While the NPCs have become extremely popular in SMLM, their usage has been modest in other techniques, and especially ExM. This is mainly due to their protein-dense nature, which is easily observed in electron microscopy. Protein-dense structures become thoroughly inter-connected after sample fixation, and the components are then difficult to separate during the expansion process. One solution is to use thorough protein digestion with proteinase K, to enable the expansion process to take place after the protein complexes have been destroyed. This lowers the quality of post-expansion labeling with antibodies, since many of the epitopes are removed. Alternatively, when antibody labeling is performed before expansion, the digestion removes the fluorophores, resulting in poor imaging. A second solution is to employ limited digestion, thereby taking low expansion factors into account (with the NPCs expanding less than the rest of the cell[6,9]). A third solution, offering better performance, is the use of complex protocols, as iterative ultrastructure expansion microscopy (iU-ExM), in which the sample is expanded multiple times[7]. Overall, NPC expansion suffers either from poor labeling (when thorough homogenization is used) or from poor expansion factors (when limited digestion is used), or requires difficult procedures, as iterative ExM (Supplementary Fig. 1a–g).

[1]Department of Neuro- and Sensory Physiology, University Medical Center Göttingen, Göttingen, Germany. [2]Department of Chemistry, GITAM School of Science, GITAM, Hyderabad, Telangana, India. [3]Institute for X-Ray Physics, University of Göttingen, Göttingen, Germany. [4]Department of Cellular Neurophysiology, Center for Integrative Physiology and Molecular Medicine (CIPMM), Saarland University, Homburg, Germany. [5]Center for Biostructural Imaging of Neurodegeneration, University Medical Center Göttingen, Göttingen, Germany. [6]NanoTag Biotechnologies GmbH, Göttingen, Germany. [7]Cluster of Excellence "Multiscale Bioimaging: from Molecular Machines to Networks of Excitable Cells" (MBExC), University of Göttingen, Göttingen, Germany. [8]These authors contributed equally: Rajdeep Chowdhury, Tiago Mimoso. ✉e-mail: sarah.koester@uni-goettingen.de; srizzol@gwdg.de; ali.shaib@med.uni-goettingen.de

Another popular sample for SMLM, STED, and related microscopy approaches has been the DNA origami nanoruler, whose popularity stems both from its labeling precision and commercial availability (e.g.,[10]). However, DNA strands connect into ExM gels using chemistries that differ from those used for proteins, implying that they are inherently limited as validation structures for this technology.

These observations leave the microtubule as a potential ExM nanoruler, having a simple geometry (relatively straight filaments), a constant thickness throughout the entire structure, and abundant, identical tubulin epitopes, oriented circularly, around the microtubule axis. As a large variety of tubulin antibodies are commercially available, microtubule immunostainings have been extremely popular, and have been employed from the earliest days of super-resolution imaging, as in the initial biological applications of STED[11], SMLM[12], or ExM[5].

With super-resolution entering the Ångstrom domain (e.g.,[13,14]), the popularity of the microtubules as natural nanorulers has suffered, with many of the most advanced technologies not employing them, as RESI[13] or MINFLUX, with few microtubule analyzes (e.g.,[15]). This is mainly due to the imprecisions introduced by microtubule immunostaining, since the use of primary and secondary antibodies leads to a large and poorly predictable displacement of fluorophores from their targets. Moreover, using microtubule-binding probes results in a surprisingly low number of fluorophores observed along single microtubules, probably due to cellular proteins masking the probe-binding sites (e.g.,[15,16]). These problems are also evident in ExM, where the density of the fluorophores is diluted by the third power of the expansion factor, leading to low-intensity images, which can only be compensated for by extraordinary efforts (e.g., expressing GFP-tubulin, and immunostaining with antibodies against GFP along with α- and β-tubuli[17]).

Ideally, one could rely on in vitro-assembled microtubules, in a cell-free environment, for testing modern imaging technologies. Assembling microtubules in cell-free systems has been a routine procedure for more than four decades[18] and the microtubules obtained are fully functional, being generally employed for analyzing cargo transport processes. Microtubule-binding probes would not be impeded by other proteins in vitro. Moreover, fluorescently conjugated tubulin could be employed, thereby eliminating the fluorophore displacement observed in immunostainings. However, a significant problem is that microtubules in vitro are labile, and are sensitive to imaging and/or fixation conditions. They would appear to be especially unsuitable for ExM, in which the sample needs to be anchored to a swellable gel before it can be expanded. During this process, the microtubules will normally depolymerize over time, even if they have been previously stabilized with a compound as taxol (since the taxol cannot be replenished during the anchoring and gelation process).

We analyzed this problem here, comparing and contrasting microtubule imaging, using ExM, in cells and in cell-free conditions, in vitro. We report a protocol that enables the preservation of microtubules in vitro, and their analysis in ExM. Moreover, by relying on the bulk labeling of proteins in cells subjected to a membrane extraction protocol, we can image microtubules in their cellular context, without any need for affinity probes, enabling strong fluorescent labeling, while eliminating the fluorophore displacement induced by such probes.

Overall, we conclude that microtubules can serve as ExM nano rulers, since their large apparent size (~60 nm), when immunostained in a conventional fashion, is sufficient to test many technologies (e.g., 10-fold expansion tools, with resolutions of 20–30 nm[19,20]), while the smaller and precise labeling provided by in vitro analysis or cellular extraction can be used in technologies with higher resolutions.

## Results
### Microtubule analysis in conventional immunostainings
For our initial investigations, we used X10 ExM[20,21], to determine the organization of microtubules in HeLa cultured cells. The cells were fixed and immunostained using four different monoclonal and polyclonal antibodies against alpha-tubulin followed by secondary antibodies conjugated to STAR635P. The samples were then anchored to gels using Acryloyl-X (Ac-X), and were homogenized using proteinase K[20,21]. Gel formation and expansion followed, before imaging the samples (Fig. 1). Microtubule labeling was satisfactory, as demonstrated by STED images, before and after expansion (Fig. 1a). The expected microtubule size was also obtained by imaging the expanded samples with a confocal or a STED microscope (Fig. 1a).

To enhance our resolution, we applied the one-step nanoscale expansion (ONE) microscopy protocol[22], in which we imaged the samples in a series of images of up to 2000 frames, which were then analyzed by higher-order temporal statistics, following the super-resolution radial fluctuations (SRRF) software implementation[23]. This procedure results in finally resolved images with sufficient precision to identify individual protein positions[22]. The higher imaging precision of this technique enabled us to separate the fluorescence signals stemming from the different antibodies, albeit the overall pattern of the antibody immunostaining was still evident, especially in a profile view (Fig. 1b, c).

The size of the observed structures resembled the known size of antibody-microtubule complexes, around $57.64 \pm 1.24$ nm (Fig. 2a, d), which has been repeatedly demonstrated using electron microscopy (e.g.,[24]) or fluorescence imaging (e.g.,[25]) The replacement of secondary antibodies with secondary nanobodies led to a substantial reduction of the apparent size, down to $31.74 \pm 0.59$ nm (Fig. 2c, e), in line with the substantially smaller dimensions of these molecules[26]. Fourier ring correlation (FRC)[23] analysis indicated a resolution estimated at $3.003$ nm $\pm 0.322$ (Supplementary Fig. 2a–d).

### Direct microtubule analysis through detergent extraction
To improve labeling accuracy and to reduce the linkage error, we investigated directly labeling microtubules using NHS-ester chemistry, eliminating the need for antibodies or nanobodies carrying fluorophores. In principle, NHS-ester labeling targets all reactive amines in the cell[27], resulting in an electron microscopy-like labeling of the cellular structures, after expansion[28].

Therefore, since NHS-ester is not specific to microtubules, it will also label nearby cellular proteins, which makes it difficult to identify their boundaries and calculate the expansion factor based on tubulin dimensions. To address this, we turned to detergent extraction procedures[29] which have been established for electron microscopy procedures, from negative staining[30] to metal shadowing[31]. This approach, which combines aldehyde fixation with detergent treatments, effectively removes most membrane components and cytoplasmic proteins, enabling the efficient analysis of the cytoskeletal components. We modified existing modern protocols[31], to make them more compatible with ExM approaches, as discussed in the Methods section and Table 1. To ensure robust cytoplasmic protein elimination, we increased the extraction buffer strength, while reducing glutaraldehyde (GA) treatments, to enable accurate fixation, but improve the subsequent gel anchoring process. This is necessary since both aldehydes and anchoring molecules primarily target amines. Extracted cells were then anchored overnight with Ac-X, before commencing gelation procedures (Fig. 3a). After expansion, we labeled the extracted cells with NHS-ester fluorescein, which conjugates itself onto the N-termini of the peptides resulting from the homogenization (protein digestion) procedure. As a control, we labeled non-expanded, extracted cells with NHS-ester, and compared them to expanded cells (Fig. 3b). Objects closely resembling cytoskeletal elements were evident in both expanded and non-expanded cells. We imaged these cellular components with increasing resolution, as shown in Fig. 3c. The first panel depicts an overview of a 3D-depth map of X10ht, followed by a 3D-deconvolved-ExSTED (dExSTED) image from the same region (See Supplementary Fig. 3a–c, for deconvolution parameters). Image decorrelation analysis indicates an achieved resolution of $7.6 \pm 0.1633$ nm (Supplementary Fig. 3d). When analyzed at higher resolution, using ONE microscopy, the filaments closely resemble the expected microtubule patterns (Fig. 3d). This impression was confirmed by line scan analyzes averaging to profiles of $24.98 \pm 0.4255$ nm (28 cross sections), in

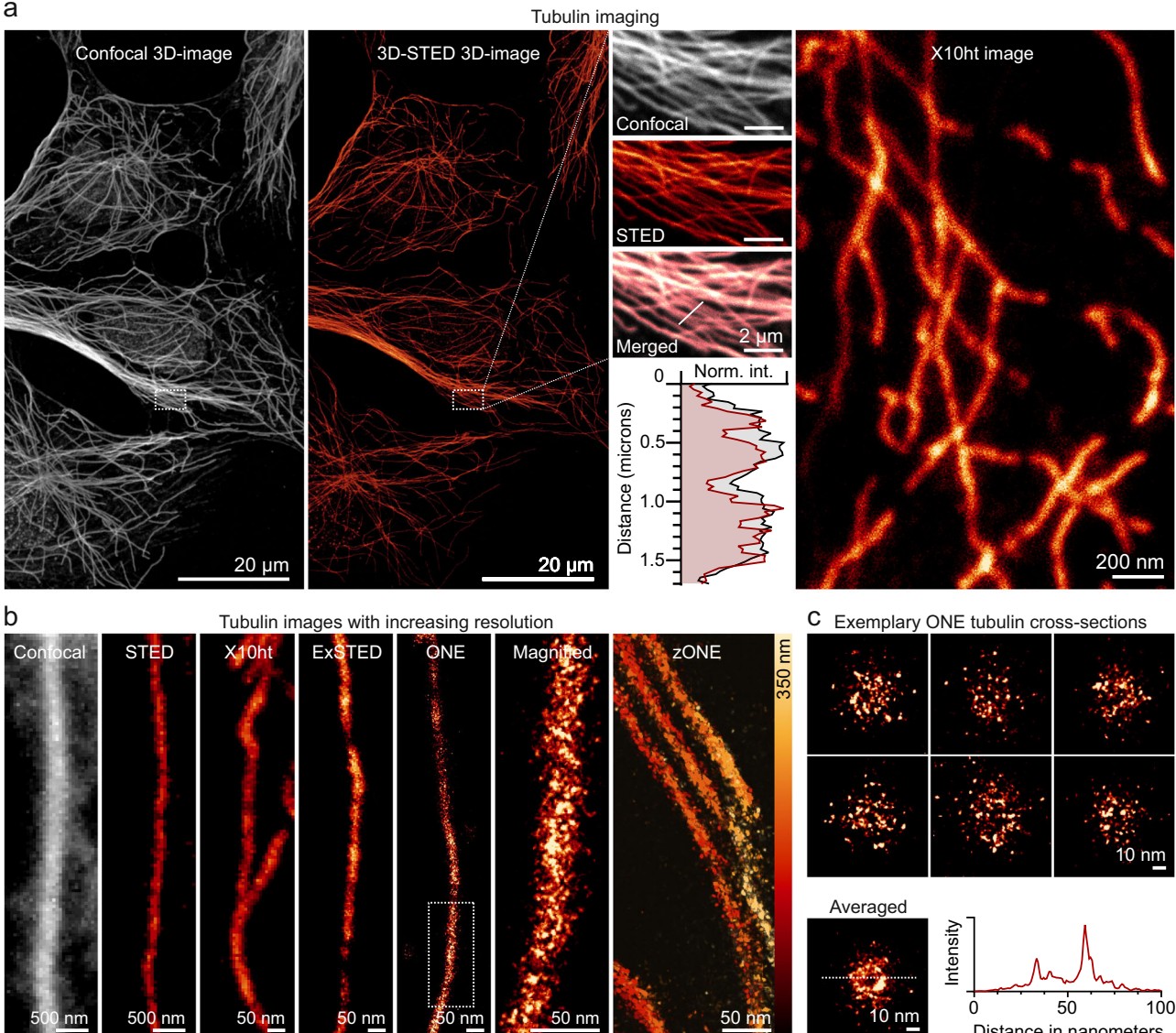

**Fig. 1 | Analysis of microtubules by conventional immunostaining in cells.**
**a**, **b**, Tubulin immunostainings (relying on primary and secondary antibodies) imaged using confocal microscope, STED, and confocal X10ht (X10 expansion, with homogenization performed by heat treatments[47]) (**a**). **b** A magnified microtubule section imaged before expansion with confocal and STED, and after expansion with confocal (X10ht), ExSTED[17], and ONE microscopy; followed by a magnified region from ONE image to show a detailed view of primary-secondary antibody labeling of tubulin. The last panel depicts ONE microscopy image at different Z-axis levels, obtained by confocal scanning at different heights (zONE). **c** Exemplary ONE tubulin cross-sections, followed by an averaged ONE image of 36 cross-sections. The graph depicts the line scans indicated by the dashed lines. The image (AVG ONE) shows an average of 28 cross-sections.

line with previous data from X-ray and CryoEM. Microtubules were readily distinguishable from smaller cytoskeletal elements labeled with NHS-ester, such as actin filaments, as previously described[22] (Supplementary Fig. 4a–c).

**Microtubule analysis in vitro, in the absence of cells**
In principle, an even higher precision should be obtained if microtubules are prepared in vitro since this avoids confusing signals corresponding to other cellular components. However, microtubules are subject to dynamic instability and have a natural tendency to depolymerize in unfavorable conditions, e.g., low temperature or the lack of GTP. This is a significant problem for ExM, as the samples need to be naturally stable to survive the processes of anchoring and gel formation.

In cellular experiments, the samples can be preserved with simple fixatives, such as formaldehyde, or even with mixtures of fixatives and acrylamide, which initialize the gel anchoring during fixation[32]. The microtubules are sufficiently stable in the cellular environment, to resist this

process, albeit modern works still recommend the use of rapid dehydration, using cold methanol, for the preservation of microtubules (e.g.,[33]) since the slow pace of aldehyde fixation allows for a certain level of depolymerization.

In vitro synthesized microtubules were even less stable than in cells, in our hands. In simple terms, two opposite processes need to be combined: first, the microtubules need to be stabilized, by the addition of taxol or its derivatives. Second, they are fixed chemically. Third, they are bound to anchor molecules that will enable their integration in ExM gels. Fourth, the gels are polymerized. This is a slow process (hours), during which taxol cannot be replaced, leading to its slow unbinding from the microtubules (since the fixation and anchoring buffers lack taxol). During fixation, tubulin molecules may unfold, also contributing to de-polymerization. Another problem is that both aldehyde fixatives and Acryloyl-X, a widely-used anchor molecule, react with primary amines on lysines. If the microtubules are thoroughly fixed, using high amounts of a rapidly-acting alde-hyde, as GA, they then barely link to the gel, since the fixative modifies most

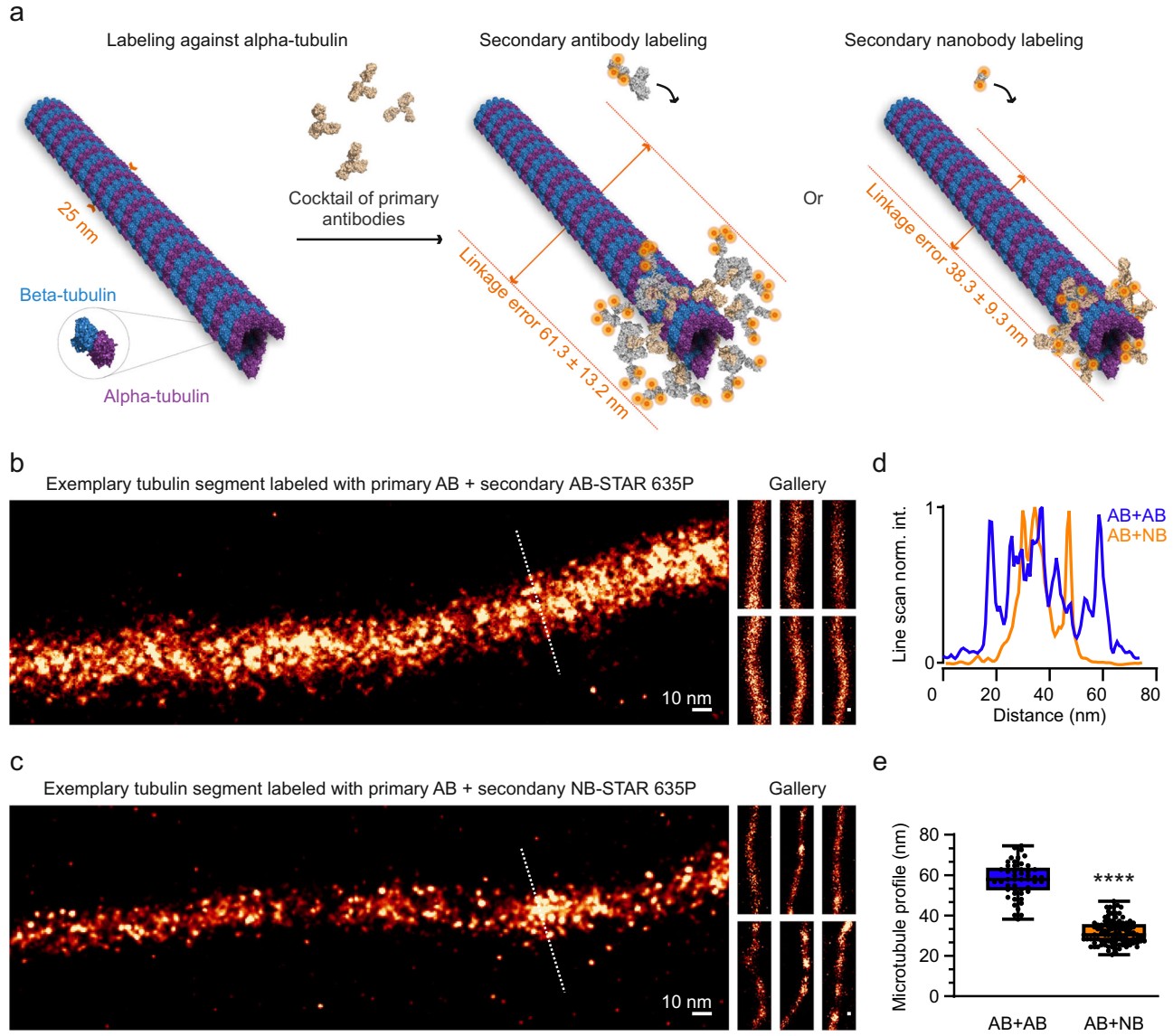

**Fig. 2 | Tubulin immunostainings with secondary antibodies or nanobodies. a** A cartoon depicting the tubulin structure which is reconstructed from PDB 1TUB with a diameter of 25 nm, followed by labeling with primaries against tubulin dimers together with either secondary antibodies with a linkage error ranging between 46 and 79 nm, or secondary nanobodies with an estimated linkage error of 30 to 47 nm. An analysis of tubulin, following immunostainings relying on a cocktail of four polyclonal primary antibodies detected using polyclonal Star635P-conjugated secondary antibodies (**b**) or one primary antibody premixed with nanobodies conjugated with STAR635P (**c**). Using a mixture of multiple primary antibodies labeled with secondary antibodies carrying the same fluorophores enhances the homogeneity of the labeling process and increases coverage, as more epitopes are decorated with antibody complexes. **d** Microtubule profile line scans obtained from the dotted lines in panels (**b**, **c**). **e** Measurements of apparent width microtubules labeled with AB + AB or AB + NB, displayed as box plot showing the medians and the 25th percentile and the range values. The width of AB − AB labeling ranged between 38.22 and 74.48 nm, with a coefficient of variation equal to 15.09%; for AB − NB labeling, the range was between 20.58 and 47.04 nm, with a coefficient of variation equal to 18.69%. $N_{AB + AB} = 49$ and $N_{AB + NB} = 101$ measurements from 2 independent experiments and at least 4 gels, two-tailed non-parametric Mann–Whitney test was applied, $p < 0.0001****$.

of the primary amines, before the addition of the anchor molecule. If they are poorly fixed, for example with the slow-acting PFA[13], they link into the gel, but only in part, since a high degree of microtubule depolymerization is observed, leading to a destruction of the structure, quite visible both before and after expansion (Fig. 4a, b and Supplementary Figs. 5, 6 and 7).

To solve these problems, we tested different chemical fixation protocols. An account of all these protocols is provided in the Methods section. In short, we maintained a delicate balance between the extent of fixation and anchors, to enable adequate fixation, while enabling the anchor molecules to still find unreacted primary lysines. We observed that only 8% PFA was able to maintain this balance, enabling us to succeeded in obtaining sufficiently accurate views of microtubules. After gel anchoring, the samples were homogenized using the homogenization protocols described in the Methods section, before expansion, and the tubulin peptides generated by the homogenization, which remained anchored into the gel, were labeled using NHS-ester chemistry, relying on fluorescein[22]. This enabled us to visualize the microtubules directly and to analyze their overall organization. Despite the obvious structural damage, due to depolymerization (Fig. 4b, c), the microtubules are still recognizable, and their profiles average to the expected size (Fig. 4d).

An unexpected benefit of this procedure is that abundant free tubulin dimers result from the depolymerization of the microtubules (Fig. 4e and Supplementary Fig. 8a–c). Their images can be averaged, to showcase an object that is similar to the known organization of this dimer (Fig. 4f).

**Table 1 | Extraction protocol solutions and commentary**

| Stock solutions | Components | Notes |
|---|---|---|
| PIPES 200 mM in 500 ml | 30.24 gm dissolve in 450 ml, add 1 M KOH until it dissolves, add water to make up the volume to 500 ml, adjust the pH to 6.9. | Aliquots: 4 °C for short-term and −20 °C for long-term storage. |
| MgCl$_2$ 10 mM in 500 ml | Dissolve 0.476 gm powder to 500 ml water. | Aliquots: 4 °C for short-term and −20 °C for long-term storage. |
| EGTA 10 mM in 200 ml | Dissolve 0.76 gm powder to 200 ml water. | Aliquots: 4 °C for short-term and −20 °C for long-term storage. |
| PEM buffer | Mix 100 mM PIPES with 1 mM MgCl$_2$, and 1 mM EGTA. | Stored for 1 month at 4 °C. |
| Extraction solution 1 | 1% Triton X-100 added to PEM buffer in the presence of 4% polyethyleneglycol (PEG) (MW 20,000–40,000), 2 μM phalloidin-Alexa633, and 2 μM taxol (paclitaxel). | PEG, phalloidin and taxol serve as stabilizers of cytoskeletal elements. Extraction solution 1 is stored for 3 days at 4 °C. This solution is unsuitable for the experiment, as excessive proteins remained attached to the cytoskeletal elements. |
| Extraction solution 2 | 1.5% Triton X-100 added to PEM buffer in the presence of 4% polyethyleneglycol (PEG) (MW 20,000–40,000), 2 μM phalloidin-Alexa633, and 2 μM taxol (paclitaxel). | Extraction solution 2 is stored for 3 days at 4 °C. This solution was sub-optimal for the experiment goal. |
| Extraction solution 3 | 2% Triton X-100 added to PEM buffer in the presence of 4% polyethyleneglycol (PEG) (MW 20,000–40,000), 2 μM phalloidin-Alexa633, and 2 μM taxol (paclitaxel). | Extraction solution 3 is stored for 3 days at 4 °C. This solution was preferred for the experiment, as it allowed observation of isolated *cellular* cytoskeletal elements. |
| Fixation solution | 1% EM grade GA in 0.1 M PEM, pH 7.3. | Fixation solution is stored for up to 7 days at 4 °C. For ExM, we recommend using 0.5-1% EM grade GA for fixation, as higher concentrations (like the ideal 2% or higher) can interfere with anchoring processes. |

## Discussion

Our results indicate that microtubules can be imaged efficiently, using ExM methods, both in cells and in cell-free conditions. At lower resolutions, immunostained microtubules appear to be continuously labeled (Fig. 1), but sparser fluorescent signals are evident at higher resolution, indicating an insufficient coverage of tubulin epitopes. The direct labeling of microtubules, using NHS ester chemistry, reveals their molecules with a lower linkage error, albeit problems with partial microtubule depolymerization still occur. These problems are solved by detergent extraction, with this approach being the simplest and most reliable procedure in employing microtubules as nanorulers.

In principle, nanorulers need to present a precise size and to have a reliable and reproducible fluorophore distribution. These purposes are optimally served by DNA origamis[10,34], which can place fluorophores at chosen positions in a rigid structure, thereby generating well-defined distances between them. The fact that most origamis are based on nucleic acid molecules makes them less attractive for ExM, where the vast majority of the applications focus on integrating proteins into the gels, rather than nucleic acids (which can, nevertheless, be targeted, using glycidyl methacrylate (GMA)[35]. Most protocols for cellular homogenization are designed to either break (e.g., by proteinase K digestion) or separate proteins from each other[8], again calling for protein-based nanorulers for such technologies. Protein-based assemblies have been used for a decade (e.g.,[36,37]), but remain challenging, especially due to difficulties in designing new interactions among proteins, while preventing numerous weak, cooperative, unwanted binding events, which are much more common for proteins than for nucleic acid molecules.

Even if nanorulers can be successfully produced, either from DNA or protein building blocks, it is difficult to introduce them into cells, so that they can be then subjected to the same process of expansion as the rest of the cellular materials, thereby serving as calibration units for the respective cells. Therefore, native cellular components would be preferable. As the artificially-produced nanorulers, they need to have a stereotypic size, along which fluorophores that can be placed at precise locations. Microtubules do fulfill the first condition, having a rigid and precisely organized geometry, but present difficulties regarding the fluorophore placement. Antibody labeling is insufficient for this, since the antibodies are placed at unpredictable locations, depending on the density of proteins in the immediate vicinity of the microtubules. The large size of the antibodies is mostly responsible for this problem, since they penetrate the meshwork of fixed proteins in cells much more poorly than smaller probes (e.g.,[38,39]). The use of secondary antibodies adds another level of uncertainty to the fluorophore placement, which is only partially alleviated by the use of secondary nanobodies (Fig. 2). Ideally, nanobodies that directly recognize tubulin should be employed, but they also have difficulties in identifying sufficient epitopes, resulting in sparsely labeled microtubules[25].

An alternative approach would be to label the microtubules fluorescently, relying on genetically-encoded probes. Transient expression of GFP-tubulin, however, results in sparse labeling of microtubules, with many of the molecules remaining in the unmodified, native form. A GFP-tubulin knock-in is substantially more successful in this regard (e.g.,[40]) but the high abundance of fluorescent, soluble tubulin results in difficult imaging conditions and sub-optimal results. Click chemistry approaches, in which unnatural amino acids have been incorporated into tubulin, enabling the conjugation of fluorophores directly on the surface of the molecules, in a site-specific fashion, should be substantially more successful, but still results in relatively inhomogeneous labeling, even in microtubules polymerized in vitro using only modified tubulin[41]. Direct chemical labeling, using fluorophores conjugated to tubulin, is only feasible for cell-free work and may interfere with microtubule polymerization if the fluorescently-conjugated tubulin molecules unfold and/or change their structures. If only a minor fraction of the molecules are conjugated fluorescently, to enable accurate polymerization, then the resulting images will again place the fluorophores in variable locations, resulting in sub-optimal images.

Ideally, all molecules would be labeled fluorescently, mimicking the density-imaging approaches of electron microscopy. This approach, entailing the use of aggressive chemical conjugation methods, as NHS-ester chemistry[8,22], which target multiple amines on each protein (at the risk of modifying the proteins too strongly, and causing them to unfold), functions well for isolated proteins, and also for cells, enabling the observation of microtubules within centrioles[28], but the high density of proteins within the cytosol implies that the resulting contrast will not be sufficient to enable the visualization of microtubules at all locations. Detergent extraction solves this issue, as demonstrated in Fig. 4.

Overall, we conclude that the description of microtubules in ExM is a feasible procedure for the validation of the different protocol steps, from homogenization and expansion to imaging. At the same time, accurate analyzes of microtubules via ExM would be extremely valuable in themselves, enabling the structural analysis of rare events, as, for example, the presence of actin filaments within the microtubule lumen[42].

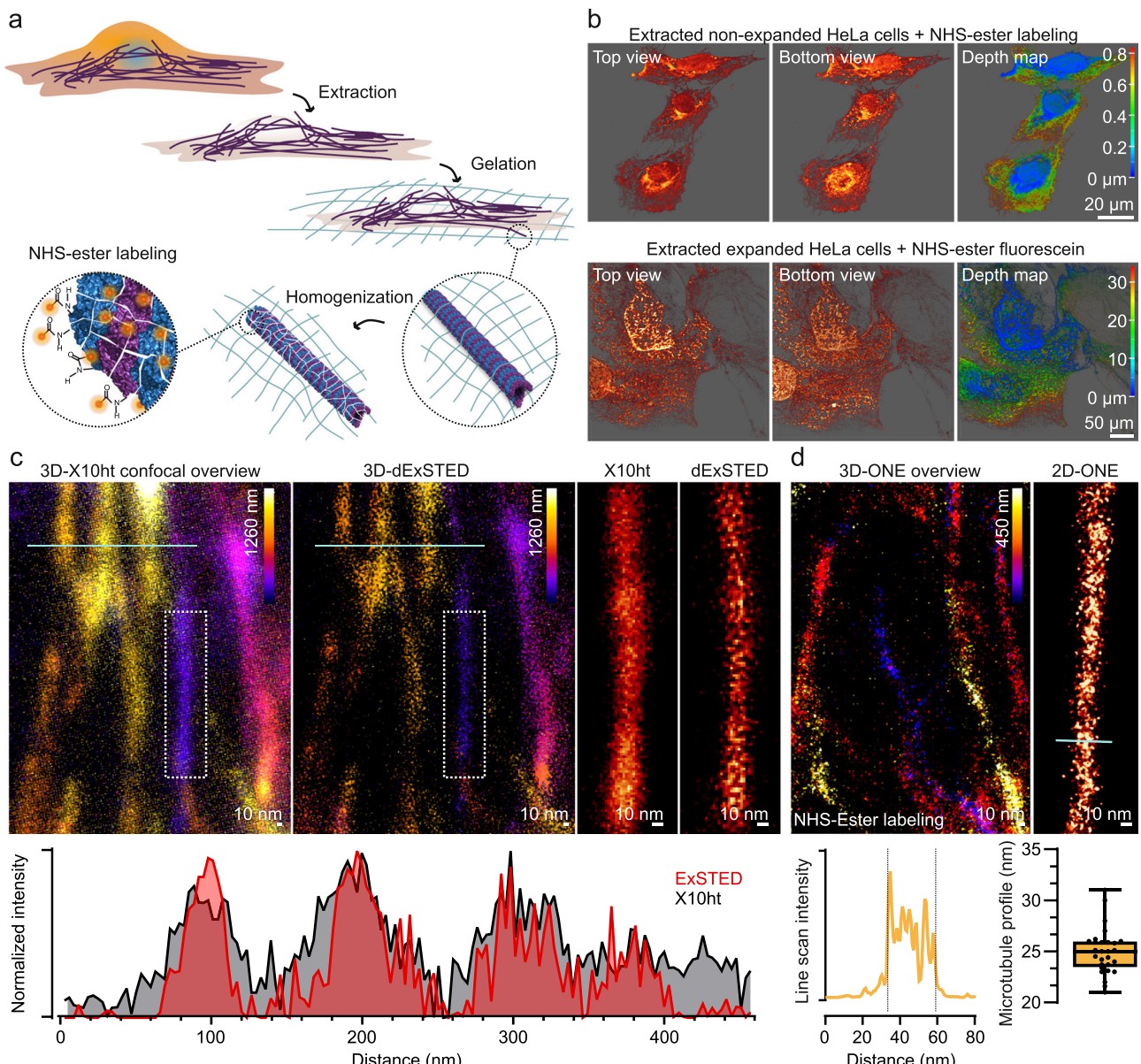

**Fig. 3 | ONE imaging of in vitro-extracted microtubules labeled with NHS-ester chemistry. a** HeLa cells were extracted using 2% triton X-100 in PEM buffer (100 mM PIPES, pH 6.9, 1 mM $MgCl_2$, and 1 mM EGTA) with phalloidin and taxol, serving as microtubule stabilizers. The extracted cells were then fixed using 1% EM-grade GA in 0.1% PEM at a final pH of 7.3. Finally, the samples were expanded and directly labeled using NHS-ester chemistry (STAR35P or fluorescein). **b** Exemplary 3D images of HeLa cells, both non-expanded (upper panel) and expanded (lower panel), labeled with NHS-ester STAR635P in top, bottom, and depth views. For STED and ExSTED imaging, we labeled the specimens with NHS-ester STAR635P due to its superior photostability and higher quantum yield, which enhances performance in STED imaging. **c** Overviews of 3D-X10ht overview image and a raw 3D-

ExSTED overview images, followed by a magnified region indicated by the dotted area of X10ht and deconvolved-ExSTED (dExSTED) The graph shows a normalized intensity line scan profile from confocal and ExSTED images. **d** 3D-ONE overview image followed by a higher resolution magnified image. The first graph shows a line scan intensity profile and the second graph shows a box plot with the median and the 25th percentile and the range values of the apparent microtubule width in nanometers. The width of NHS-ester labeled microtubules ranged from a minimum of 21.00 to a maximum of 31.00 nm and a coefficient of variation equal to 9.012%. $N = 28$ microtubule segments, acquired from 2 independent experiments from 4 gels.

## Materials and methods
### Cell culture for microtubule immunostaining
U-2 OS cell line (CLS #300364, Eppelheim, Germany) were grown in Dulbecco's Modified Eagle Medium (DMEM Merck #D5671, Darmstadt, Germany) supplemented with 10% FCS (fetal calf serum, Merck #S0615), 4 mM glutamine (ThermoFisher Scientific #25030-024, Waltham, USA) and 1% penicillin-streptomycin (ThermoFisher Scientific #15140148). For imaging purposes, freshly split cells were grown overnight on poly-L-lysine coated coverslips (Merck #P2658) which reduced detachment during immunostaining.

### Microtubule immunostaining
To extract lipid membranes, U-2 OS cells were first incubated for 1 min with 0.2% saponin (Sigma–Aldrich #47036) in cytoskeleton buffer, consisting of 10 mM MES (Merck #M3671), 138 mM KCl, 3 mM $MgCl_2$, 2 mM EGTA and 320 mM sucrose, at pH 6.1. The cells were then fixed using 4% PFA and 0.1% GA (PanReac #A3166, Darmstadt, Germany), in the same buffer. Unreacted aldehydes were quenched by incubating with 0.1% $NaBH_4$ (Sigma–Aldrich #71320), for 7 min in PBS, followed by a second quenching step with 0.1 M glycine (Carl Roth #3187) in PBS for 10 min.

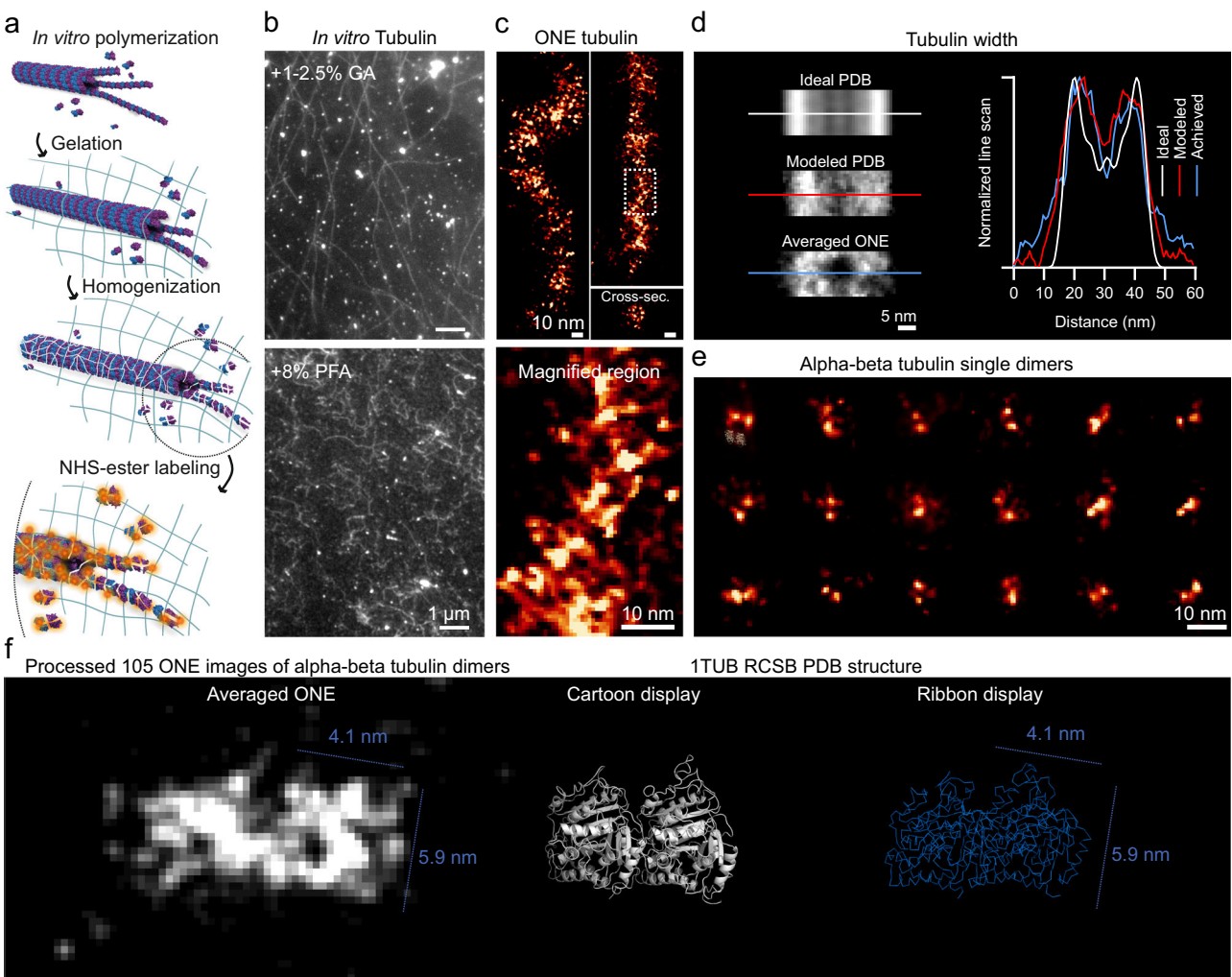

**Fig. 4 | ONE imaging of in vitro assembled microtubules. a, b** The upper panel shows in vitro-assembled microtubules that are stably fixed with 1-2.5% GA. GA above 0.2% interfered with anchoring into the gels. The lower panel shows less stable microtubules, fixed with 8% PFA. These microtubules are deformed and tend to depolymerize, but this fixation does allow a reasonable degree of anchoring to the gels. **c** ONE images of microtubules fixed with 8% PFA and expanded and labeled using NHS-ester fluorescein. The lower panel shows a magnified region. **d** An averaged analysis of side views of microtubule segments. The top panel shows an ideal image, obtained by convoluting the PDB structure of a microtubule segment (3J2U) with a fluorescent PSF, in which every amino acid is labeled fluorescently. The second panel shows a realistic model, in which sparser labeling is considered,

and in which different microtubule segments are overlaid with slight tilt angle differences (up to 5°). The third panel shows an averaged processed ONE image of 175 partially depolymerized tubulin segments, acquired from N = 3 independent experiments from at least 5 gels. The graph shows the respective line scans across each of the panels. **e** A gallery of alpha-beta tubulin dimers that were left unpolymerized. **f** The first panel shows a tubulin dimer reconstructed from 105 dimers, performed by overlapping the individual molecule images. The second panel shows a 1TUB PDB cartoon structure, and the third panel shows a ribbon display of the molecules. A simple comparison indicates that although the averaged ONE image protein shape lacks structural details, the 2D-dimensions are consistent with the resolved ground truth depicted in the cartoon displays.

The samples were blocked and simultaneously permeabilized using 2% BSA and 0.1% Triton X-100 (Sigma–Aldrich #9036-19-5), in PBS for 30 min at room temperature. For immunostaining, a mixture of primary tubulin antibodies (Sigma–Aldrich #T6199, Synaptic Systems #302211, Göttingen, Germany, Synaptic Systems #302203, Abcam #ab18251, Cambridge, UK) with either secondary antibodies (Abberior #ST635P-1001, Göttingen, Germany) or secondary nanobodies (in separate experiments; NanoTag Biotechnologies GmbH #N1202 and #N2402, Göttingen, Germany) in blocking buffer were prepared using a ratio of 1:5 for the primary:secondary probes, respectively. The antibody mixture was left at room temperature for 30 min to ensure that the complex formation is complete. Afterwards, the cells were incubated with the antibody mixture for 1 h followed by washing with permeabilization buffer (5 buffer exchanges, each one for 10 min) and three washes with PBS, before continuing with cellular expansion.

**Detergent extraction**

The extraction procedure, based on a modified protocol version from Svitkina, 2009[31], was optimized for ExM protocols. HeLa cells were plated on Poly-L-Lysin (PLL)-free coverslips. Poly-lysine aggregates are anchored and then labeled in amine-reactive chemistries, so they should be avoided in extraction-ExM procedures. Upon reaching confluency, the medium was removed, and cells were rinsed with pre-warmed PBS at 37 °C. The PBS was replaced with extraction buffer and incubated for 7 min at 37 °C. Cells were then rinsed three times with PEM, at room temperature, and fixed with fixation solution for 20 min at room temperature. For details on buffers, solutions, and optimization notes, refer to Table 1.

**Preparation of in vitro microtubule samples**

The preparation of in vitro assembled microtubules was carried out following protocol[43]. Unlabeled tubulin (TebuBio #T240-A, Offenbach,

**Table 2 | Summary of the different tested conditions for expanding in vitro synthesized microtubules**

| Fixative used | Level of fixation | Anchoring material | Level of anchoring |
|---|---|---|---|
| 3% PFA | Doesn't work, mostly tubulin dimers | Acryloyl-X | Anchored to the gel |
| 4% PFA | Doesn't work, mostly tubulin dimers | Acryloyl-X | Anchored to the gel |
| 8% PFA | Worked, although very long filaments are broken | Acryloyl-X | Anchored to the gel, ONE data obtained from this condition |
| 16% PFA | Worked, although very long filaments are broken, better than 8% PFA | Acryloyl-X | Doesn't anchored to the gel |
| 3% PFA + 0.25% GA | Doesn't work, mostly tubulin dimes with very few small broken filaments | Acryloyl-X | Doesn't anchor to the gel |
| 3% PFA + 0.5% GA | Partially worked, mostly small broken filaments | Acryloyl-X | Doesn't anchor to the gel |
| 0.5% GA | Worked, although very long filaments are broken | Acryloyl-X | Doesn't anchor to the gel |
| 0.5% GA | Worked, although very long filaments are broken | GMA | Doesn't anchor to the gel |
| >1% GA | Preserved the filaments at optimal level | Acryloyl-X | Doesn't anchor to the gel |

Germany) was reconstituted to a concentration of 100 μM. Stabilized microtubules were prepared by polymerizing through step-wise increase of the tubulin concentration. Initially, a 3 μM tubulin solution in M2B (magnesium 2X buffer: 80 mM PIPES with 1 mM EGTA and 2 mM MgCl$_2$, pH 6.8, adjusted with KOH), in the presence of 1 mM GMPCPP (Jena Bioscience #NU-405L, Jena, Germany) was prepared at 37 °C to nucleate short microtubule seeds. Next, the total tubulin concentration was increased to 9 μM in order to grow long microtubules. To avoid further microtubule nucleation, we added 1 μM tubulin at a time from a 42 μM stock solution and waited for 15 min between the successive steps. We centrifuged the polymerized microtubules for 10 min at 13,000 × g to remove any non-polymerized tubulin and short microtubules. We discarded the supernatant and carefully resuspended the pellet in 800 μl M2B-taxol (M2B supplemented with 10 μM taxol (Merck #T7402, Darmstadt, Germany)).

**Fixation of in vitro microtubules**
ExM protocols entail the anchoring of molecules to gels, a process that takes a significant time interval (hours). To maintain the accurate organization of the proteins, they need to be chemically fixed to each other. Even if microtubules are stabilized using taxol, the tubulin molecules still need to be conjugated to each other, to avoid the destabilization of the molecular assemblies during anchoring and gelation. We therefore fixed the microtubule by implementing a chemical fixation method. We attempted to fix the microtubules immediately after they were assembled. After resuspension of the microtubule pellet in taxol, we added the fixatives. We first tested the most common fixative, PFA, applied at different concentrations (3% to 16%) to the microtubules and incubated for 30 min at RT. Although 3–4% PFA preserves most the biological structures, it does not preserve the structure of microtubules at all. It is a slow fixative, which first modifies single molecules, and then later, slowly, conjugates them to each other. The modification of the single molecules may cause them to unfold and leave the microtubule structure, thereby explaining the depolymerization observed here and in other works relying on PFA. This prompted us to try higher concentration of PFA. We found that 8% PFA preserve the microtubule to some extent, presumably by speeding up the process of conjugating molecules to each other. As 8% PFA showed some promise, we further increased the PFA concentration to 16%, which worked better than 8% PFA.

Overall, we observed that PFA fixation failed to preserve the long filaments. For this reason, we tested faster fixatives, as combinations of PFA and GA (see Table 2). We also tested GA alone (0.25% to 2.5%), by incubating the microtubule suspension with GA for 10 min at room temperature. Although we found a GA concentration which preserves the microtubule structure optimally, unfortunately, GA-stabilized microtubules do not anchor to the gel.

**Immobilization of in vitro microtubules on the coverslip**
Fixatives were found to interfere strongly with the anchoring process. Therefore, it is necessary to wash off the fixative from the sample before the anchoring. Further, processing a sample for expansion microscopy involves substantial washing steps. Hence, some kind of attachment (covalent or non-covalent) of the sample to the coverslip is a prerequisite. Several strategies have been attempted. First, the sample was put onto the coverslip and then the coverslip was allowed to semi-dry at RT under a biosafety cabinet. As PFA and GA are volatile, they should evaporate during the drying process, leaving behind the microtubules onto the coverslip. Unfortunately, this process resulted in complete disassembly of microtubules, probably due to changes in the buffer concentration, during drying. A similar result was obtained by quick heating (30 s) of the coverslip at 37 °C, using a heat block to get rid of the volatile fixatives. Finally, we discovered a strategy which is described below.

We placed 10 μl of microtubule solution at the center of the coverslip followed by covering it with a piece of parafilm so that sample was spread over the entire area of the coverslip, due to gravity and capillarity forces. We left it for 10 min and then we gradually removed the parafilm using a tweezer. Microtubules remained firmly attached with the glass surface of the coverslip presumably due to electrostatic interactions. Fixatives were then removed by washing the coverslip with PBS.

Although 1% GA or above preserved the microtubule morphology to an optimal level, it inhibits the anchoring and hence, is not used for the ONE protocol. Conversely, and as previously described[44], 4% PFA is too slow to fix microtubules, which tend to depolymerize. Glutaraldehyde acts much faster, stabilizing the long tubules and preserving their morphology (Fig. 4b). However, glutaraldehyde is highly efficient and modifies all available amines. Since Acryloyl-X, an amine-reacting molecule, is used for anchoring to ExM gels, this leads to extremely inefficient anchoring in vitro, resulting in poor outcomes. In living cells, fixation is less aggressive than in vitro because glutaraldehyde reacts with a plethora of small metabolites, such as amino acids, partially quenching its activity. This partial quenching enables superior anchoring in cells, resulting in better outcomes. For in vitro assembled microtubules, we found that 8% PFA allowed both structure preservation and anchoring reasonably well and hence, we proceeded further with this condition for ONE imaging (See Table 1).

**X10 expansion procedures**
For most of the samples, we used NHS-ester chemistry to anchor proteins into the gels, using Acryloyl-X SE (Thermo Fisher Scientific, #A-20770)[20,21]. Coverslips were first washed 2x with PBS (pH 7.4) and then incubated overnight at 4 °C with 0.3 mg/ml Acryloyl-X in PBS. For one of the samples, we employed GMA based anchoring[35] which involves incubation of the sample containing coverslip with 0.04% GMA in 100 mM bicarbonate buffer (pH 8.5) for 3 h at room temperature.

The samples were then washed with PBS (3x, 5 min each) while preparing the gel monomer solution, as described earlier[22]. Eighty microliters of monomer solution were placed on parafilm and were covered by upside-down coverslips containing samples. Monomer solution was allowed to polymerize overnight at room temperature, in a humidified chamber. Homogenization of microtubule (in vitro) was performed by incubating the gel with 8 U/ml proteinase K (Merck, #P4850) in digestion buffer (800 mM guanidine HCl, 2 mM CaCl₂, 0.5% Triton X-100, in 50 mM TRIS, Merck, #8382J008706), overnight at 37 °C. On the other hand, in vivo, microtubule was homogenized by autoclaving for 60 min at 110 °C in disruption buffer (5% Triton-X and 1% SDS in 100 mM TRIS, pH 8.0) followed by 90 min of waiting time at room temperature to cool down. In vitro, microtubule containing gel was labeled using 20-fold molar excess of NHS-ester fluorescein (ThermoFisher Scientific, #46409,) in sodium bicarbonate buffer at pH = 8 for 1 h. This step is not applied to already immuno-stained in vivo microtubules. Gel was finally expanded by repeated ddH₂O washing, for several hours until the gel is fully expanded. A fully expanded gel is 15–17 cm in diameter and kept in a chamber with water until we are ready for imaging.

## Ultrastructure expansion microscopy (U-ExM) procedure
PFA-fixed HEK cells were expanded using U-ExM as previously described[7,45,46]. Cells were incubated in an anchoring solution with 1.4% FA and 2% AA in 1x PBS at 37 °C for 3 h. An 80 μl gelation solution containing 19% SA, 10% acylamide (AA), and 0.1% BIS in 1x PBS with TEMED and APS was placed on a parafilm-lined ice-cooled humidified chamber, and 18 mm coverslips with cultured cells were inverted on it. The setup was left on ice for 5 min and incubated at 37 °C for 1 h. Homogenization was performed using a heat-detergent method in a denaturation buffer (200 mM SDS, 200 mM NaCl, 50 mM Tris-BASE, pH 9) at 95 °C for 1.5 h. Gels were washed twice in 1x PBS and then multiple times in ddH2O for full expansion. Prior to staining, the gels were shrunk back by placing them in 1x PBS. Primary antibodies against NUP98-NUP96 polyclonal rabbit antibody (Proteintech, 12329-1-AP and Proteintech, 24439-1-AP) and secondary nanobodies (FluoTag-X2 anti-Rabbit Star635P IgG, NanoTag, N2402-AB635P-S and FluoTag-X2 anti-Rabbit ATTO643 IgG, NanoTag, N2402-At643-S) solutions were prepared using 2% BSA in 1x PBS and incubated at 37 °C with continuous agitation. The gels were incubated with primary antibodies for 2 h and 30 min, followed by three washes of 15 min each with 0.1% Tween-20 in 1x PBS. Subsequently, the gels were incubated with secondary nanobodies for 1 h and 30 min and washed four times for 30 min each with 0.1% Tween-20 in 1x PBS. Finally, two additional washes were performed in PBS. Gels were washed multiple times with ddH₂O until fully expanded.

## Expansion factor determination
Determining the expansion factor in ExM is crucial for several reasons, including correcting scale bars for the expansion factor and ensuring accurate and consistent imaging results. Several factors influence the physical expansion of biological specimens: (1) cross-linker concentration, (2) homogenization method, (3) sample thickness, (4) hydration time, and (5) polymerization conditions. As detailed in our previous publication[47], changes in these parameters lead to changes in the expansion factor. For instance, lowering the cross-linker concentration generally results in greater expansion factors, but makes the gels more fragile and difficult to handle. Different homogenization protocols affect the uniformity and extent of expansion. In DMAA-based gel recipes introduced in 2018[20], a heat-alkaline disruption yields an expansion factor of 8X to 9X for cytoskeletal elements[47], while proteinase K yields a 10X expansion factor for cellular components[22,47], and 13X to 15X for purified protein molecules[22]. The expansion factor can rise to approximately 20X, as recently reported[48], when the polymer recipe and oxygen purging are optimized. Another important factor is the duration and hydration of gels in their expansion procedure. ddH₂O washes are necessary until the gel reaches its final expected size. Lastly, variations in polymerization conditions, such as chemical components, temperature, oxygen purging, and

time, impact the final expansion factor. To achieve the desired expansion and quality of imaging in expansion microscopy, these elements must be carefully controlled and optimized. Even slight variations in any of these steps may alter the final expansion factor. **It is important to measure the gel dimensions and record the apparent expansion factor of the gel. In addition, it is important to measure organelle landmarks within cells—particularly the cell nucleus—to estimate the correct expansion factor of the cells**. Depending on the ExM protocol and the desired resolution, more precise measurements are required, especially when the achievable resolution is higher than 20 nm. Here, we propose three different methods with known linkage errors to label microtubules, as nanorulers for a more precise estimation of the expansion factor size.

## Microscope systems
To check the microtubule morphology under different conditions, we used an Olympus IX71 epi-fluorescence microscope equipped with a 100X, 1.41 NA, oil immersion objective. Images were captured using F-view II CCD camera (pixel size was 64.5 nm).

Before imaging, water was removed from the gel chamber. A rectangular piece of gel was cut and excess water was removed from the get using a soft tissue paper. Then the gel piece was placed on a self-engineered net that sits on top of a coverslip in a mounting chamber. For ONE imaging, the mounting chamber was placed in either a confocal or epifluorescence microscope.

Confocal imaging was performed on a TCS SP5 STED microscope (Leica Microsystems, Wetzlar, Germany) with confocal modality. Samples were excited with laser lines at 488, 561, and 633 nm using a HCX Plan Apochromat 100X STED objective (1.4 NA, oil immersion), and the emissions were collected using an AOTF filter with pre-dedicated wavelength range on HyD detectors at a scanning speed of 8 kHz. Image sequences of 8-bit depth were acquired at a line format of 128 × 128 pixels, one pixel equals to 98 nm. The scanning modality on a confocal was set to "minimize time interval" (Leica LAS software). Line averaging during scanning was avoided to maintain natural fluctuations of fluorophores.

2D, 3D-STED, and ONE imaging imaging were performed on a STELLARIS 8 PP STED FALCON microscope (Leica Microsystems, Wetzlar, Germany). Confocal overview images were obtained using the navigator function. All imaging modalities employed an HC PL APO 100x/1.4 OIL STED W objective. The main excitation source was a White Light Laser (WLL), tuned to the optimal wavelength for each fluorophore and operating at a pulse frequency of 80 MHz. 3D-STED images were captured with a theoretical pixel size ranging from 20 to 40 nm. This process utilized STED depletion beas at 775 nm, with a repetition rate of 80 MHz and output power exceeding 1.5 W, along with a 50 nm xy Vortex Donut and a 130 nm z Donut. Detection of near-infrared and/or far-red emissions was carried out using a Power HyD R SP detector or Power HyD X SP detectors, with the appropriate STED 3X notch filter set in place. The detectors were configured for either counting intensity or counting τSTED modes. 3D reconstructions were accomplished via the 3D viewer in LAS-X. For ONE microscopy, images were captured using a 12 kHz Tandem Scanner. The theoretical pixel size was set to 92 nm, which after computation with 32-bit image depth, resulted in a final computed pixel size of 0.92 nm. Each ONE image was acquired with 1500–2000 frames per channel. It should be noted that ONE images can be obtained with fewer frames; however, this results in compromised resolution.

For epi-fluorescence imaging, sample was excited through an Andor iXon Ultra 888 100X (1.49 NA) TIRF objective attached to an Olympus IX83 TIRF microscope equipped with a LAS-VC 4-channel (405, 488, 532, and 633 nm) laser lines and emissions were collected on the Andor iXon camera. Image sequences of 16-bit depth were acquired at a line format of 485×479 pixels, one pixel equals to 130 nm.

## Image analysis and statistics
Image sequences were analyzed using a "ONE Platform" free plugin in Fiji. This plugin has multiple functionalities depending on the type of

data. For 2D ONE imaging, image sequences were analyzed using "ONE Microscopy" functionality. Details about this plugin were provided in our earlier publication[22]. In brief, image sequences were loaded into the software and analyzed using the above-mentioned plugin. There are options of providing the input parameters. All the data were processed using TRAC4 modality with radiality magnification and ring axes were set to 10 and 8, respectively. The software computes the image sequences and provides final ONE image. It may be noted that this software implements automatic drift correction in xy-plane and chromatic correction for multi-color imaging. However, since the ONE plugin does not require videos with optical sections for the analysis, there is no option for compensating the z-drift in the sample. Users must ensure their microscopes are stable, which can typically be achieved by turning on the microscopes overnight (or at least 5–6 h before starting), to ensure that all components reach the same temperature. As ONE image acquisitions usually last less than a minute, z-drifts of the microscope stage become undetectable. However, if users detect z-drifts after imaging the gel for some time, this indicates that the gel has started to shrink due to drying. At this stage, the sample focal plane would be unstable, and the expansion factor would be non-uniform within the video recording and subsequent recordings. It is advisable to stop imaging and replace the gel piece with a freshly cut one. The stability of the gel can be extended by placing small wet tissue sections next to the gel in the imaging chamber, which maintains the gel's hydrated condition for several hours. Plots and statistical analyzes were conducted using GraphPad Prism 9.3.1 (GraphPad Software), SigmaPlot 10 (Systat Software), and Matlab. Detailed statistical information is provided in the respective figure captions. Figures were crafted with CorelDraw 23.5 (Corel Corporation) and Adobe Photoshop Beta 25.4.0 release (Adobe). Additionally, select materials sourced from Envato Elements (Envato 2025) are licensed for use in this publication.

## Tubulin dimers averaged ONE analysis

The averaging analysis of tubulin dimers was performed using Matlab. An automated threshold was applied, to detect the dimers, and they were then centered manually, before being automatically rotated and aligned to each other. The peaks of fluorescence were then detected for each particle, using a band-pass filter[49]. The position of the peaks was estimated to a precision of 0.384 nm (down from 1 nm pixel sizes), and these positions were then mapped onto the image shown in the respective figure.

## Reporting summary

Further information on research design is available in the Nature Portfolio Reporting Summary linked to this article.

## Data availability

Image data are available from the corresponding authors on reasonable request. Summary data are presented in the paper and Supplementary Files. The supplementary information document contains all Supplementary Figs. (Supplementary Figs. 1–8) and Supplementary References. The source data is provided in Supplementary Data file.

## Code availability

The ONE platform plugin software (source code) is available from Zenodo (https://doi.org/10.5281/zenodo.13685267).

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

## Acknowledgements
The study was supported by grants from the German Research Foundation (Deutsche Forschungsgemeinschaft, DFG), under Germany's Excellence Strategy - EXC 2067/1- 390729940, to S.K. and S.O.R., and under SFB1286 (B02, to S.K. and S.O.R., Z04, to F.O.). Funding from the Chan Zuckerberg Initiative (CZI), grant iNano, 2023-321180(5022)GB-1586232, to S.O.R. is acknowledged. R.C. was supported by a fellowship from the Alexander von Humboldt Foundation. We acknowledge support by the Open Access Publication Funds/transformative agreements of the Göttingen University.

## Author contributions
A.H.S. and S.O.R. conceived the study. A.H.S., S.O.R., F.O., and S.K. supervised the experiments. R.C., T.M., N.K., D.K., and A.H.S. carried out the experiments. A.A.C. developed the ONE plugin. A.H.S. and S.O.R. analyzed the data. S.O.R. and A.H.S. wrote the manuscript with contribution from all the authors.

## Funding

## Competing interests
S.O.R. and F.O. are shareholders of NanoTag Biotechnologies GmbH. The remaining authors declare no competing interests.
