## [Transparent Peer Review file · Communications Biology]

Microtubules as a versatile reference standard for expansion microscopy

Corresponding Author: Professor Silvio Rizzoli

Version 0:

Reviewer comments:

Reviewer #1

(Remarks to the Author)

The manuscript by Chowdhury et al. demonstrates that microtubules in cells and ex-vitro can be used as efficient reference system to validate different expansion microscopy protocols. Indeed, references systems for expansion microscopy are rare because DNA origami that are the preferred sample for super-resolution microscopy require digestion of nucleic acids with nucleases while expansion microscopy protocols digest proteins with proteases. The data shown are interesting and some of them are also convincing but other parts of the manuscript need more explanation and controls before it can be accepted for publication.

In the introduction the authors argue that nuclear pore complexes (NPCs) are not suited as reference for expansion microscopy (ExM) because they are too protein dense. However, microtubules are also protein-dense structures and should thus suffer from the same problems. I am convinced that an optimization of labeling, fixation and digestion (as has been performed for microtubules in the present manuscript) will likewise enable the use of NPC as reference system for ExM experiments.

The authors claim that microtubules as they appear in the ONE-microscopy images are homogeneously labeled. However, the images (e.g. Fig. 1b) clearly show areas with lower labeling density. This becomes even more pronounced when labeling is performed with secondary nanobodies. Here, the filaments exhibit completely unlabeled areas (Fig. 2c). Why do IgG antibodies show more homogenous labeling?

If the method used does achieve a spatial resolution of 3 nm as claimed by the authors, do the signals shown in Figs. 1c and 2c represent individual antibodies?

Overall, I think the STED microscopy images of expanded microtubules show a similar resolution and quality as the ONE microscopy images. Can the authors comment on that?

Why did the authors use fluorescein for their NHS-labeling approach. Fluorescein is known to exhibit a low photostability. I assume that other dyes should outperform fluorescein. A comparison of the image quality dependent on different dyes would be interesting.

Did the authors simply assume an expansion factor of 10 or how did they determine the expansion factor? Maybe the expansion factor is much higher for purified ex-vitro proteins?

Fig. 4f shows the overlay of 105 One-microscopy images of tubulin dimers and 1e images of single dimers. I think the single dimer images show clearly that the resolution of the method is not sufficient to reveal structural information as indicated in Fig. 4f.

What are the bright signals in Fig. 4b?

Reviewer #2

(Remarks to the Author)

The manuscript titled "Microtubules as a versatile reference standard for expansion microscopy" proposes that microtubules should be used as reference structures for validating expansion microscopy (ExM) techniques. The authors argue that microtubules are suitable because their diameter, when immunostained, is large enough to validate resolutions better than 50 nm. They suggest using direct microtubule analysis through detergent extraction for higher precision. As expansion microscopy is getting more popular in recent years, this study should be helpful to the community. The methods described in this manuscript are detailed and the results are presented in a well-organized manner. I have the following comments for the authors.

1. In Figure 2 e and Figure 3 d, the scatterplot shows the distribution of the measured width of the microtubules. Does each of the data points correspond to a separate microtubule? If so, what is the width variance of a single microtubule? Figure 3 d does not include the population number for the scatter plot.

2. In the study by Laporte et al. [1] effect of Paraformaldehyde (PFA) and Glutaraldehyde (GA) on microtubule integrity has been mentioned. It is observed that PFA tends to disrupt the integrity of the U2OS microtubule more than PFA+GA fixation. Is there a reason for the authors to observe a different trend in their experiments in the section “Fixation of in vitro microtubules”?

3. The authors mentioned that the drying process to get rid of fixative and attaching the cells to the coverslip resulted in disassembly of the microtubules. There has been mention of attaching cells to glass substrate without air drying by silane-PEG coated [2] slides, centrifuging cells to slides after silanization and aldehyde wash [3], or using poly-L-lysine [4]. Do the authors believe bonding the sample to the substrate prior to drying might be helpful for the experiment?

4. Although the time duration of the confocal scanning is not mentioned, was there any change in the expanded gel due to evaporation of the ddH₂O? or there was no evaporation?

5. The authors mentioned that the z-drift was not compensated during the image analysis. Was the observed z-drift insignificant? Does it have any effect on the accuracy of the results?

References:

1. Klimas, A., Gallagher, B.R., Wijesekara, P. et al. (2023). Magnify is a universal molecular anchoring strategy for expansion microscopy. *Nat Biotechnol* 41, 858–869
2. Cuveillier, C., Saoudi, Y., Arnal, I. and Delphin, C. (2021). Imaging Microtubules in vitro at High Resolution while Preserving their Structure. *Bio-protocol* 11(7): e3968
3. Maples, J. A. (1985). A Method for the Covalent Attachment of Cells to Glass Slides for Use in Immunohistochemical Assays. *American Journal of Clinical Pathology*, 83(3), 356–363
4. Rodig, S. J. (2020). Attaching Suspension Cells to Slides for Staining. *Cold Spring Harbor Protocols*, 2020(12)

Version 1:

Reviewer comments:

Reviewer #1

(Remarks to the Author)

The authors carefully addressed all my concerns in the revision.

Therefore, I thank the authors for their efforts and recommend publication of the manuscript in its revised form.

Reviewer #2

(Remarks to the Author)

Thank you to the authors for addressing the comments. There are no further comments.

Microtubules as a versatile reference standard for expansion microscopy

Rajdeep Chowdhury^{1,2*}, Tiago Mimoso^{3*}, Abed Alrahman Chouaib⁴, Nikolaos Mougios^{1,5}, Donatus Krah¹, Felipe Opazo^{1,5,6}, Sarah Köster^{3,7§}, Silvio O. Rizzoli^{1,5,7§}, Ali H. Shaib^{1§}

¹Institute for Neuro- and Sensory Physiology, University Medical Center Göttingen, Göttingen, Germany.

²Department of Chemistry, GITAM School of Science, GITAM, Hyderabad 502329, Telangana, India.

³Institute for X-Ray Physics, University of Göttingen, Göttingen, Germany.

⁴Department of Cellular Neurophysiology, Center for Integrative Physiology and Molecular Medicine (CIPMM), Saarland University, Homburg, Germany.

⁵Center for Biostructural Imaging of Neurodegeneration, University Medical Center Göttingen, Göttingen, Germany.

⁶NanoTag Biotechnologies GmbH, Rudolf-Wissell-Straße 28a, 37079 Göttingen, Germany.

⁷Cluster of Excellence "Multiscale Bioimaging: from Molecular Machines to Networks of Excitable Cells" (MBExC), University of Göttingen, Göttingen, Germany.

*equal first authors, alphabetically.

§corresponding authors, alphabetically

Replies to the Reviewer comments

Please find below our replies to all of the comments of the Reviewers.

The original comments are shown in *italics*, with our replies in regular font.

Revisions in the manuscript are marked in **light blue** font for clarity.

Reviewer #1

The manuscript by Chowdhury et al. demonstrates that microtubules in cells and ex-vitro can be used as efficient reference system to validate different expansion microscopy protocols. Indeed, references systems for expansion microscopy are rare because DNA origami that are the preferred sample for super-resolution microscopy require digestion of nucleic acids with nucleases while expansion microscopy protocols digest proteins with proteases. The data shown are interesting and some of them are also convincing but other parts of the manuscript need more explanation and controls before it can be accepted for publication.

We thank the Reviewer for the valuable comments and feedback on our manuscript. Each point has been carefully addressed, leading to significant improvements in the quality of our work.

In the introduction the authors argue that nuclear pore complexes (NPCs) are not suited as reference for expansion microscopy (ExM) because they are too protein dense. However, microtubules are also protein-dense structures and should thus suffer from the same problems. I am convinced that an optimization of labeling, fixation and digestion (as has been performed for microtubules in the present manuscript) will likewise enable the use of NPC as reference system for ExM experiments.

We agree with the Reviewer that this is an important point. To address it, we now include a series of additional experiments on NPCs, expanded using different gel types. The results are included below, in Review Fig. 1. We also include a detailed explanation in the manuscript.

Overall, we conclude that the use of NPCs in ExM requires optimization, and that their proper visualization in ExM probably requires more expertise in sample handling than microtubules.

Review Figure 1. ExM imaging of nuclear pore complexes. **a, b**, non-expanded NPC overview and a representative magnified nucleus shown in confocal and 2D-STED acquisitions. **c**, A representative 3D-overview of HEK cells that were expanded according to the U-ExM protocol described in Gambarotto *et al.*, 2021¹ and were post-expansion labelled against NUP98 and NUP205 with primary rabbit antibodies coupled to secondary anti-rabbit nanobodies carrying STAR635P fluophores. The gels were then labelled with NHS-ester fluorescein, to show the general cellular morphology. **d**, U-ExM ONE microscopy images of NPCs from the designated area shown in **(c)**. **e**, Similar overview but for X10 expansion, reaching an expansion factor of 8.5. A similar labeling procedure was performed, prior to expansion. The two lower panels show two representative NPCs from the upper image. **f**, A random averaged alignment of NPCs (left panel), followed by an 8-fold symmetry average alignment (right panel). **g**, A circle profile that shows the distribution of the NUPs in the right panel of **(f)**. The averaging procedure was performed as described in Aktalay *et al.*, 2023², for MINFLUX imaging of NPCs. The resulting data are in very good agreement with the MINFLUX results, especially considering the signal blurring induced by the use of polyclonal antibodies in our experiments, rather than a NUP96-HaloTags in MINFLUX.

The authors claim that microtubules as they appear in the ONE-microscopy images are homogeneously labeled. However, the images (e.g. Fig. 1b) clearly show areas with lower labeling density. This becomes even more pronounced when labeling is performed with secondary nanobodies. Here, the filaments exhibit completely unlabeled areas (Fig. 2c). Why do IgG antibodies show more homogenous labeling?

The Reviewer is right in noting this point, which is due to using different sets of antibodies for the two stainings. In Fig. 2c, we only used one primary antibody, while in Fig. 1b we used a cocktail of four different primary antibodies. This results in a very different coverage. We have now explained this in the phrasing of the respective figure legend.

The legend of Figure 2 is amended as follows:

“Figure 2. Tubulin immunostainings with secondary antibodies or nanobodies...

... b & c, An analysis of tubulin, following immunostainings relying on a cocktail of four polyclonal primary antibodies detected using polyclonal Star635P-conjugated secondary antibodies (b) or one primary antibody premixed with nanobodies conjugated with STAR635P (c). Using a mixture of multiple primary antibodies labeled with secondary antibodies carrying the same fluorophores enhances the homogeneity of the labeling process and increases coverage, as more epitopes are decorated with antibody complexes...”

If the method used does achieve a spatial resolution of 3 nm as claimed by the authors, do the signals shown in Figs. 1c and 2c represent individual antibodies?

According to our previous experiments with antibodies (Shaib *et al.*, 2024, <https://www.nature.com/articles/s41587-024-02431-9>), these signals represent groups of fluorophores on individual antibodies. In Fig. 2c, it may be possible to observe single fluorophores present on X2 secondary nanobodies, in regions labelled at lower densities. Please see Fig. 2a, Sup. Fig 10a&b, Sup. Fig. 14a-e, Sup. Fig. 15 and Sup. Fig. 16 in Shaib *et al.*, 2024, for a detailed description of images obtained with antibodies and antibody fragments.

Overall, I think the STED microscopy images of expanded microtubules show a similar resolution and quality as the ONE microscopy images. Can the authors comment on that?

We thank the Reviewer for highlighting this point. Indeed, the deconvolved STED microscopy images of expanded microtubules appear similar to the ONE microscopy images. We have now included a decorrelation analysis³ to estimate the achieved resolution in dExSTED, as presented in Supplementary Figure 3 of the manuscript and in **Review Fig. 2**. This resolution analysis indicates that dExSTED achieves a resolution ranging from 7 to 8 nm, which is approximately 2-fold poorer than the one achieved by ONE microscopy in the same samples, ranging from 2.5 to 4.3 nm, as shown in Supplementary Figure 2.

Review Figure 2. Resolution analysis of dExSTED images. The first panel shows a decorrelation analysis³ plot of all decorrelation functions computed for estimating image resolution. The second box plot shows the individual values calculated from 10 dExSTED images acquired from 10 different cells from two independent experiments.

Why did the authors use fluorescein for their NHS-labeling approach. Fluorescein is known to exhibit a low photostability. I assume that other dyes should outperform fluorescein. A comparison of the image quality dependent on different dyes would be interesting.

We thank the Reviewer for pointing out this aspect. We apologize for not including all the necessary information on sample labelling in **Fig. 3** caption. We now correct this, and we now explain that the samples and gels were labelled with NHS-ester STAR 635P when using STED and ExSTED imaging. As the Reviewer correctly mentions, photostable dyes outperform fluorescein in photon-demanding imaging methods like STED. This is now corrected as follows:

“Figure 3. ONE imaging of *in vitro*-extracted microtubules labelled with NHS-ester chemistry. **a**, HeLa cells were extracted using 2% triton X-100 in PEM buffer (100 mM PIPER, pH 6.9, 1 mM MgCl₂, and 1 mM EGTA) with phalloidin and taxol, serving as microtubule stabilizers. The extracted cells were then fixed using 1% EM-grade glutaraldehyde in 0.1% PEM at a final pH of 7.3. Finally, the samples were expanded and directly labeled using NHS-ester chemistry (STAR35P or fluorescein). **b**, Exemplary 3D images of HeLa cells, both non-expanded (upper panel) and expanded (lower panel), labeled with NHS-ester STAR635P in top, bottom, and depth views. For STED and ExSTED imaging, we labeled the specimens with NHS-ester STAR635P due to its superior photostability and higher quantum yield, which enhances performance in STED imaging. **c**, Overviews of 3D-X10ht overview image and a raw 3D-ExSTED overview images, followed by a magnified region indicated by the dotted area of X10ht and deconvolved-ExSTED (dExSTED) The graph shows a normalized intensity line scan profile from confocal and ExSTED images. **d**, 3D-ONE overview image followed by a higher resolution magnified region. The first graph shows a line scan intensity profile and the second graph shows a box plot with the median and the 25th percentile and the range values of the apparent microtubule width in nanometers. The width of NHS-ester labelled microtubules varied from a minimum of 21.00 to a maximum of 31.00 nm and a coefficient of variance equal to 9.012%. N = 28 microtubule segments, acquired from 2 independent experiments from 4 gels.”

However, we did use NHS-ester fluorescein for confocal and ONE imaging as two arguments speak in favour of fluorescein:

- Its price, which is, per mg, 100-fold cheaper than modern dyes. This argument is important, since large gels need to be incubated with excess NHS-ester dyes, in volumes of many millilitres.
- We compared NHS-fluorescein with NHS-Star635P, as shown in the **Supplementary Fig. 10**, below, from our previous bioRxiv communication on ONE microscopy. Using substantially more stable dyes does not necessarily lead to superior results when imaging with ONE microscopy. This is due to ONE microscopy's reliance on imaging parameters that use a fast resonant scanner (8000 Hz) and an imaging window of about

40 to 60 seconds with low laser intensities, where fluorescein bleaching is less pronounced compared to classical imaging frequencies at 200 Hz. (please see a comparison of different dyes in **Sup. Fig. 3** in Shaib *et al.*, 2024).

bioRxiv preprint doi: <https://doi.org/10.1101/2022.08.03.502284>; this version posted March 10, 2023. The copyright holder for this preprint (which was not certified by peer review) is the author/funder, who has granted bioRxiv a license to display the preprint in perpetuity. It is made available under aCC-BY-NC-ND 4.0 International license.

Supplementary Fig. 10. GABA_A receptor and otoferlin galleries. a, An overview of images of GABA_A receptors. b, The images display GABA_A receptors in different 3D positions. The positional indications are best guesses performed by an experienced investigator. c, Overview images of otoferlin (right panel), and blank buffer as a control (left panel). d, Otoferlin labelled with NHS-ester fluorescein ONE images in different 3D positions. e, Otoferlin labelled with NHS-ester STAR635P ONE images.

This Figure is acquired from our preprint on bioRxiv, can be found here: <https://www.biorxiv.org/content/10.1101/2022.08.03.502284v2>, as Sup. Fig 10. Panel d & e shows comparisons of a purified protein molecules labelled with NHS-ester fluorescein and NHS-ester STAR 635P, respectively.

To answer the Reviewer's comment thoroughly, we analyzed HeLa cells that were extracted and then expanded and labelled with NHS-ester STAR 635P. As shown in **Review Fig. 3**, the results were similar to those from cells labelled with NHS-ester fluorescein.

Review Figure 3. ONE imaging of extracted cells. The samples were expanded using the X10 protocol, and the gels were labeled with NHS-ester STAR635P. The resulting ONE images closely resemble those labeled with NHS-ester fluorescein.

Did the authors simply assume an expansion factor of 10 or how did they determine the expansion factor?

We have now added a section on expansion factor determination in the Methods section (line 486), to address this point for both our Reviewer and the readers of the manuscript:

“Expansion factor determination

Determining the expansion factor in ExM is crucial for several reasons, including correcting scale bars for the expansion factor and ensuring accurate and consistent imaging results. Several factors influence the physical expansion of biological specimens: 1) cross-linker concentration, 2) homogenization method, 3) sample thickness, 4) hydration time, and 5) polymerization conditions. As detailed in our previous publication⁴, changes in these parameters lead to changes in the expansion factor. For instance, lowering the cross-linker concentration generally results in greater expansion factors, but makes the gels more fragile and difficult to handle. Different homogenization protocols affect the uniformity and extent of expansion. In DMAA-based gel recipes introduced in 2018⁵, a heat-alkaline disruption yields an expansion factor of 8X to 9X for cytoskeletal elements⁴, while proteinase K yields a 10X expansion factor for cellular components^{4, 6}, and 13X to 15X for purified protein molecules⁶. The expansion factor can rise to approximately 20X, as recently reported⁷, when the polymer recipe and oxygen purging are optimized. Another important factor is the duration and hydration of gels in their expansion procedure. ddH₂O washes are necessary until the gel reaches its final expected size. Lastly, variations in polymerization conditions, such as chemical components, temperature, oxygen purging and time, impact the final expansion factor. To achieve the desired expansion and quality of imaging in expansion microscopy, these elements must be carefully controlled and optimized. Even slight variations in any of these steps may alter the final expansion factor. **It is important to measure the gel dimensions and record the apparent expansion factor of the gel. In addition, it is important to measure organelle landmarks within cells—particularly the cell nucleus—to estimate the correct expansion factor of the cells.** Depending on the ExM protocol and the desired resolution, more precise measurements are required, especially when the achievable resolution is higher than 20 nm. Here, we propose three different methods with known linkage errors to label microtubules, as nanorulers for a more precise estimation of the expansion factor size.”

Maybe the expansion factor is much higher for purified ex-vitro proteins?

Yes, this is indeed the case. This is now addressed in the additional paragraph mentioned above.

Fig. 4f shows the overlay of 105 One-microscopy images of tubulin dimers and 1e images of single dimers. I think the single dimer images show clearly that the resolution of the method is not sufficient to reveal structural information as indicated in Fig. 4f.

We agree with the Reviewer, we need to differentiate here between the resolution of the method itself, which is sufficient to reveal at least some structural information (Shaib *et al.*, 2024, Fig. 3a-c and Sup. Fig. 22)⁶, and the resolution obtained in **Fig. 4f** in the current manuscript, in which we only present a hand-driven overlay of 2D images of tubulin dimers. This overlay is not intended to present structural information. A thorough structural analysis would require a substantially larger number of images and the application of artificial intelligence-based software, as described in Shaib *et al.*, 2024, which accounts for the varying

3D positions of molecules. In this manuscript, **Fig. 4f** is included solely to demonstrate that the 2D images of the molecules are consistent with their expected size and shape. However, the Reviewer is correct in noting this potential implication, and we have now revised the text to ensure it does not hint or imply such an interpretation. Below is the amended figure legend (light blue):

“Figure 4. ONE imaging of *in vitro* assembled microtubules. a, b, The upper panel shows *in vitro*-assembled microtubules that are stably fixed with 1-2.5% glutaraldehyde (GA). GA above 0.2% interfered with anchoring into the gels. The lower panel shows less stable microtubules, fixed with 8% PFA. These microtubules are deformed and tend to depolymerize, but this fixation does allow a reasonable degree of anchoring to the gels. **c,** ONE images of microtubules fixed with 8% PFA and expanded and labelled using NHS-ester fluorescein. The lower panel shows a magnified region. **d,** An averaged analysis of side views of microtubule segments. The top panel shows an ideal image, obtained by convoluting the PDB structure of a microtubule segment (3J2U) with a fluorescent PSF, in which every amino acid is labelled fluorescently. The second panel shows a realistic model, in which sparser labelling is considered, and in which different microtubule segments are overlaid with slight tilt angle differences (up to 5°). The third panel shows an averaged processed ONE image of 175 partially depolymerized tubulin segments. The graph shows the respective line scans across each of the panels. **e,** A gallery of alpha-beta tubulin dimers that were left unpolymerized. **f,** The first panel shows a tubulin dimer reconstructed from 105 dimers, performed by overlapping the individual molecule images. The second panel shows a 1TUB PDB cartoon structure, and the third panel shows a ribbon display of the molecules. *A simple comparison indicates that although the averaged ONE image protein shape lacks structural details, the 2D-dimensions are consistent with the resolved ground truth depicted in the cartoon displays.*”

What are the bright signals in Fig. 4b?

These are “clumps” of tubulin molecules, induced by fixation *in vitro*. The more aggressive the fixative, the faster such assemblies form, as observed by a comparison between glutaraldehyde and PFA.

Reviewer #2

The manuscript titled "Microtubules as a versatile reference standard for expansion microscopy" proposes that microtubules should be used as reference structures for validating expansion microscopy (ExM) techniques. The authors argue that microtubules are suitable because their diameter, when immunostained, is large enough to validate resolutions better than 50 nm. They suggest using direct microtubule analysis through detergent extraction for higher precision. As expansion microscopy is getting more popular in recent years, this study should be helpful to the community. The methods described in this manuscript are detailed and the results are presented in a well-organized manner. I have the following comments for the authors.

We thank the Reviewer for the valuable comments and feedback on our manuscript. Each point has been carefully addressed, leading to significant improvements in the quality of our work.

1. In Figure 2 e and Figure 3 d, the scatterplot shows the distribution of the measured width of the microtubules. Does each of the data points correspond to a separate microtubule? If so, what is the width variance of a single microtubule? Figure 3 d does not include the population number for the scatter plot.

The scatter plot in Figure 2e shows the mixed distribution of measured widths of different microtubule segments, as well as regions within single microtubule segments. Some of the microtubules were quite long, and we therefore moved along them in different imaging windows. The width ranged between 38.22 nm and 74.48 nm, in microtubules labelled using primary and secondary antibodies (AB-AB labelling), with a coefficient of variation equal to 15.09%.

For microtubules labelled using primary antibodies and secondary nanobodies (AB-NB labelling), the values ranged between 20.58 nm and 47.04 nm, with a coefficient of variation equal to 18.69%.

The scatter plot of Figure 3d shows line scans measurements from separate microtubules. The population was indeed not included, for which we apologize. The information was now made available: N = 28, from 4 gels, generated in 2 independent experiments. The width of NHS-ester labelled microtubules varied from a minimum of 21.00 to a maximum of 31.00 nm and a coefficient of variation equal to 9.012%. We now add this information to the respective figures.

We also now revise all the Experimental Ns in the respective figures in the manuscript, and we present more details on the statistics of the conducted experiments, as shown below:

Figure 2: e, Measurements of apparent width microtubules labeled with AB+AB or AB+NB, displayed as box plot showing the medians and the 25th percentile and the range values. The width of AB-AB labelling ranged between 38.22 and 74.48 nm, with a coefficient of variation equal to 15.09%; for AB-NB labelling, the range was between 20.58 and 47.04 nm, with a coefficient of variation equal to 18.69%. $N_{AB+AB} = 49$ and $N_{AB+NB} = 101$ measurements from 2 independent experiments and at least 4 gels, two-tailed non-parametric Mann Whitney test was applied, $p < 0.0001$ ****.

Figure 3: d, 3D-ONE overview image followed by a higher resolution magnified image. The first graph shows a line scan intensity profile and the second graph shows a box plot with the median and the 25th percentile and the range values of the apparent microtubule width in nanometers. The width of NHS-ester labelled microtubules ranged from a minimum of 21.00 to a maximum of 31.00 nm and a coefficient of variation equal to 9.012%. N = 28 microtubule segments, acquired from 2 independent experiments from 4 gels.

Figure 4: d, An averaged analysis of side views of microtubule segments. The top panel shows an ideal image, obtained by convoluting the PDB structure of a microtubule segment (3J2U) with a fluorescent PSF, in which every amino acid is labelled fluorescently. The second panel shows a realistic model, in which sparser labelling is considered, and in which different microtubule segments are overlaid with slight tilt angle differences (up to 5°). The third panel shows an averaged processed ONE image of 175 partially depolymerized tubulin segments, acquired from N = 3 independent experiments from at least 5 gels. The graph shows the respective line scans across each of the panels. **e**, A gallery of alpha-beta tubulin dimers that were left unpolymerized.

Sup. Fig. 2: c & d, the graphs show minimal and average FRC in nm with their SEM, N = 5 measurements from 2 independent experiments.

Sup. Fig. 3: c, A box plot graph depicting the average measured gel-glass refractive index (RI) measured from four gels of 4 independent experiments, using Abbe-Refractometer AR4 (Kruss Optronic GmbH, Germany).

Sup. Fig. 4: a, The first panel shows an ONE overview image where extracted cytoskeletal elements of varying widths are typically observed in at least 4 independent experiments. The second two panels show two magnified regions, with cross sections that fit the profiles of actin and tubulin structures, as shown in the adjacent graph in (b). **c**, An exemplary gallery of actin images, next to an exemplary tubulin segment, for comparison.

Sup. Fig. 8: b, Fourier ring correlation analysis. The first-left panel displays an exemplary FRC of ONE tubulin image with a pixel size of 98 nm. The bottom-left panel shows ONE image overlaid over FRC map using screen-blend mode. A magnified region featuring individual tubulin molecule represented in split and screen-blended modes. **c**, The graphs show box plots with the medians and the 25th percentile and the range values of Minimal and average calculated FRC, N = 8 images from 2 independent experiments.

2. In the study by Laporte et al. [1] effect of Paraformaldehyde (PFA) and Glutaraldehyde (GA) on microtubule integrity has been mentioned. It is observed that PFA tends to disrupt the integrity of the U2OS microtubule more than PFA+GA fixation. Is there a reason for the authors to observe a different trend in their experiments in the section “Fixation of *in vitro* microtubules”?

We assume that the Reviewer refers to the following publication:

Visualizing the native cellular organization by coupling cryofixation with expansion microscopy (Cryo-ExM). Laporte MH, Klena N, Hamel V, Guichard P. Nat Methods. 2022 Feb;19(2):216-222. doi: 10.1038/s41592-021-01356-4.

We fully agree with the results of this work, which is also in agreement with our previous analyses (e.g. Richter et al., EMBO Journal, 2018)⁸. When whole cells are used, PFA is too slow to fix microtubules, which tend to depolymerize. Glutaraldehyde is much faster, and stabilizes the tubules. *In vitro*, glutaraldehyde remains fast, and stabilizes long, straight microtubules (Fig. 4b). However, since nothing else is being fixed, glutaraldehyde is, too efficient, and modifies all of the available amines. As we use an amine-reacting molecule, Acryloyl-X, for anchoring to ExM gels, this implies that the anchoring process will be extremely inefficient when using glutaraldehyde *in vitro*, leading to poor results. In living cells, the fixation is less aggressive than *in vitro*, because glutaraldehyde will react with a plethora of small metabolites, as amino acids, resulting in its partial quenching. This enables the superior anchoring obtained in cells, and the subsequent results.

We have now included this information in the Methods section, as shown below:

“Although 1% GA or above preserved the microtubule morphology to an optimal level, it inhibits the anchoring and hence, not used for the ONE protocol. Conversely, and as previously described⁸, 4% PFA is too slow to fix microtubules, which tend to depolymerize. Glutaraldehyde acts much faster, stabilizing the long tubules and preserving their morphology (Fig. 4b). However, glutaraldehyde is highly efficient and modifies all available amines. Since Acryloyl-X, an amine-reacting molecule, is used for anchoring to ExM gels, this leads to extremely inefficient anchoring *in vitro*, resulting in poor outcomes. In living cells, fixation is less aggressive than *in vitro* because glutaraldehyde reacts with a plethora of small metabolites, such as amino acids, partially quenching its activity. This partial quenching enables superior anchoring in cells, resulting in better outcomes. For *in vitro* assembled microtubules, we found that 8% PFA allowed both structure preservation and anchoring reasonably well and hence, we proceeded further with this condition for ONE imaging (See Table 1).”

3. The authors mentioned that the drying process to get rid of fixative and attaching the cells to the coverslip resulted in disassembly of the microtubules. There has been mention of attaching cells to glass substrate without air drying by silane-PEG coated [2] slides, centrifuging cells to slides after silanization and aldehyde wash [3], or using poly-L-lysine [4]. Do the authors believe bonding the sample to the substrate prior to drying might be helpful for the experiment?

The semi-drying process was indeed attempted for microtubules prepared *in vitro* and not for classical cell specimens, this resulted in the disassembly of the microtubules.

The Reviewer notes several important conditions, which we discuss below:

- The use of silane-PEG coated slides⁹. Here the protocol requires the use of microtubules generated from a mixture of unmodified and biotinylated tubulin, which are anchored to the silane-PEG-biotin molecules by the use of neutravidin (which binds biotin both on the microtubules and on the silane-PEG, simultaneously). This procedure would make our analysis difficult, since the NHS-ester dyes would label both neutravidin and the microtubules. These molecules would be difficult to differentiate, in the one-color labeling. As the size of neutravidin is not negligible (around 60 KDa), this procedure would not be optimal for our approaches.
- The use of aldehyde washes would destroy more amines¹⁰ on the microtubules, before gel anchoring, so it would be counter-productive, as explained for the previous comment.
- The use of poly-L-lysine^{11, 12} also results in the presence of many peptides (composed of lysine) in the immediate vicinity of the microtubules. As for neutravidin, they will be labeled by the NHS-ester dyes, implying that confusing images will be obtained.

Coating the coverslips with bovine serum albumin (BSA) to let the microtubules attach to them suffers the same consequences, as the NHS-ester labelling would also stain BSA resulting a “carpet” fluorescent signal. This led us to use the described immobilization procedure, which worked well for our purposes.

4. Although the time duration of the confocal scanning is not mentioned, was there any change in the expanded gel due to evaporation of the ddH₂O? or there was no evaporation?

In our experience, X10 and DMAA-based gel variants are more resistant to drying compared to other ExM protocols, and depending on the microscope room conditions, they are usually stable for about 2 hours. Then indeed they start drying to a level where shrinking happens and an xy- and z movement is apparent to the naked eye. Usually, at this stage, we recommend replacing the current gel piece with a new one. However, users can also increase the lifespan of the gel by adding wet tissues in the imaging chamber, beside the gel. This usually makes the gel piece stable for several hours. These details are now mentioned in the text in the Image analysis segment (please see below for point 5).

5. The authors mentioned that the z-drift was not compensated during the image analysis. Was the observed z-drift insignificant? Does it have any effect on the accuracy of the results?

Indeed, the ONE plugin does not offer z-drift correction, because the ONE software analysis does not require multiple optical sections to function. In most of our images, there are no xyc(z)t sections, rather xyt or xyct, and thus the z-drift correction is impractical. In Shaib *et al.*, 2024, we meticulously described how to reduce gel drifts using a specially designed chamber, and then how to accurately correct and test the results (See Sup. Figures 2, 7, 8 & 30 and optimization section in Methods).

Nonetheless, users will need to ensure the stability of their imaging microscopes in the longitudinal z-direction. Most of the images presented in this manuscript were generated using a Leica confocal STED SP5 microscope, purchased in 2007. Like other confocal systems, it requires approximately 5 hours to stabilize, as there is a recorded 60-micron z-drift during this period. This drift becomes undetectable afterward. Given that ONE imaging is relatively fast, mostly not exceeding 1 minute, there is practically no z-drift detectable in our ONE imaging or ExSTED imaging.

However, if z-drift occurs, it is typically an indication of gels drying and shrinking. In this case, as the sample focal plane is unstable and the gel expansion factor is not uniform within the

video recording and subsequent recordings, users should stop imaging the respective gels. We have now added these details to the Image Analysis section in the Methods, as shown below (light blue):

“Image analysis

Image sequences were analysed using a “ONE Platform” free plugin in Fiji. This plugin has multiple functionalities depending on the type of data. For 2d ONE imaging, image sequences were analysed using “ONE Microscopy” functionality. Details about this plugin were provided in our earlier publication.⁴ In brief, image sequences were loaded into the software and analysed using the above-mentioned plugin. There are options of providing the input parameters. All the data were processed using TRAC4 modality with radially magnification and ring axes were set to 10 and 8, respectively. The software computes the image sequences and provides final ONE image. It may be noted that this software implements automatic drift correction in xy-plane and chromatic correction for multi-color imaging. **However, since the ONE plugin does not require videos with optical sections for the analysis, there is no option for compensating the z-drift in the sample. Users must ensure their microscopes are stable, which can typically be achieved by turning on the microscopes overnight (or at least 5-6 hours before starting), to ensure that all components reach the same temperature. As ONE image acquisitions usually last less than a minute, z-drifts of the microscope stage become undetectable. However, if users detect z-drifts after imaging the gel for some time, this indicates that the gel has started to shrink due to drying. At this stage, the sample focal plane would be unstable, and the expansion factor would be non-uniform within the video recording and subsequent recordings. It is advisable to stop imaging and replace the gel piece with a freshly cut one. The stability of the gel can be extended by placing small wet tissue sections next to the gel in the imaging chamber, which maintains the gel’s hydrated condition for several hours.”**

References

1. Gambarotto, D., Hamel, V. & Guichard, P. Ultrastructure expansion microscopy (U-ExM). *Methods Cell Biol* **161**, 57-81 (2021).
2. Aktalay, A., Khan, T.A., Bossi, M.L., Belov, V.N. & Hell, S.W. Photoactivatable Carbo- and Silicon-Rhodamines and Their Application in MINFLUX Nanoscopy. *Angew Chem Int Ed Engl* **62**, e202302781 (2023).
3. Descloux, A., Grussmayer, K.S. & Radenovic, A. Parameter-free image resolution estimation based on decorrelation analysis. *Nat Methods* **16**, 918-924 (2019).
4. Saal, K.A. et al. Heat denaturation enables multicolor X10-STED microscopy. *Sci Rep* **13**, 5366 (2023).
5. Truckenbrodt, S. et al. X10 expansion microscopy enables 25-nm resolution on conventional microscopes. *EMBO Rep* **19** (2018).
6. Shaib, A.H. et al. One-step nanoscale expansion microscopy reveals individual protein shapes. *Nat Biotechnol* (2024).
7. Wang, S. et al. Single-shot 20-fold expansion microscopy. *Nat Methods* (2024).
8. Richter, K.N. et al. Glyoxal as an alternative fixative to formaldehyde in immunostaining and super-resolution microscopy. *EMBO J* **37**, 139-159 (2018).
9. Cuveillier, C., Saoudi, Y., Arnal, I. & Delphin, C. Imaging Microtubules in vitro at High Resolution while Preserving their Structure. *Bio Protoc* **11**, e3968 (2021).
10. Maples, J.A. A method for the covalent attachment of cells to glass slides for use in immunohistochemical assays. *Am J Clin Pathol* **83**, 356-363 (1985).

11. Klimas, A. et al. Magnify is a universal molecular anchoring strategy for expansion microscopy. *Nat Biotechnol* **41**, 858-869 (2023).
12. Rodig, S.J. Attaching Suspension Cells to Slides for Staining. *Cold Spring Harb Protoc* **2020** (2020).